# Load transfer mechanism and critical length of anchorage zone for anchor bolt

**Xingliang Xu[1,2], Suchuan Tian[1]***

**1** Key Laboratory of Deep Coal Resource Mining, Ministry of Education of China, China University of Mining and Technology, Xuzhou, Jiangsu, China, **2** Mining Department, Xinjiang Institute of Engineering, Urumqi, Xinjiang Uygur Autonomous Region, China

* tiansc@cumt.edu.cn

**Data Availability Statement:** All relevant data are within the manuscript and its Supporting Information files.

**Funding:** This work was supported, and financed, by the General Program of the National Natural Science Foundation of China (51864044).

## Abstract

The length of anchorage zone of an anchor bolt affects the distribution of axial force and shear stress therein. Based on a shear–displacement model, the load distribution of anchor bolts in the elastic deformation stage was analysed. Moreover, the mechanical response of threaded steel anchor bolts with different anchorage lengths was explored through pull-out test and numerical simulation. The results showed that axial force and shear stress were negatively exponentially distributed within the anchorage zone of anchor bolts in which there were the maximum axial force and shear stress at the beginning of the anchorage zone. In the elastic deformation stage of the anchorage, the longer the anchorage length, the more uniformly the shear stress was distributed within the anchorage zone and the larger the ultimate shear stress; however, there was a critical anchorage length, which, when exceeded, the ultimate shear stress remained unchanged. The calculation formula for the critical anchorage length was deduced and a reasonable anchorage length determined. The research result provides an important theoretical basis for rapid design of support parameters for anchor bolts.

## Introduction

As a key parameter affecting the design of bolt supports, the length of anchorage zone influences the anchoring force and support effect of anchor bolts, however, a theoretical basis for such a design remains absent, resulting in unreasonable anchorage lengths, thus leading to anchor support failure or extra cost[1,2]. Therefore, it is a challenge to guarantee that anchorage lengths satisfy design requirements while saving cost and therefore it is necessary to explore the load transfer mechanism and reasonable anchorage length of anchor bolts.

The load transfer mechanism of anchor bolts is a research hot-spot. The shear stress on anchor surface in the pull-out process can be divided into three parts: cohesion, mechanical self-locking force, and friction force[3]. Many mechanical models have been proposed: the shear lag model for an anchoring system based on the condition of considering bonding conditions of different interfaces[4], the simple trilinear constitutive model that describe the shear slip of the bonding interface between the anchor cable and grouting body[5], the stick-slip

**Competing interests:** The authors have declared that no competing interests exist.

relationship and the trilinear stick–slip model established through pull-out tests on anchor bolts[6,7], the three-parameter and two-parameter combined-power models of the distribution of axial force within the anchorage zone[8], the hyperbolic function model of load transfer by using mathematical–mechanical methods[9]. Zhu(2009) derived a function describing the distribution of frictional resistance on anchor bolts in an elastic homogeneous rock mass[10]. By applying displacement–shear stress theory and finite element analysis (FEA), the shear stress in the anchorage zone is distributed following a Gaussian function along the anchorage length. Through various *in situ* and laboratory tests[11], the distribution characteristics of axial force within the anchorage zone was obtained[12]. Despite the aforementioned research, no consensus has been reached as to the stress distribution in the anchorage zone.

As for research on anchorage length, the failure behaviours of bonded anchorage bodies under a fixed anchorage length was explored [13,14], the bearing capacity did not significantly increase when the anchorage length exceeded the critical anchorage length[15]. Huang(2018) proposed a method for calculating the critical anchorage length of anchor bolts and verified its feasibility through engineering case studies[16]. Based on the bonding effect, The anchorage length has a serious influence on the bearing capacity of anchor bolts and shear stress on interfaces under the effect of cyclic load[17–19]. The calculation formula for the critical anchorage length of anchor bolts can be deduced according to the principle of displacement compatibility between the anchorage body and surrounding rock[20–22]. Liu(2010) thought that the anchorage length has to exceed 20 times the diameter of the anchor bolt when applying full-thread GFRP anchor bolts in situ[23]. The aforementioned research achievements remain mostly hypothetical, and do not take the design requirements of actual parameters of anchor bolts into account.

In the present study, the mechanical properties and stress distribution characteristics of the anchorage zone under different anchorage lengths were explored to reveal the load transfer mechanism of the anchorage zone and propose a method for designing a reasonable anchorage length of anchor bolts.

## Analysis of mechanical properties of the anchorage zone

An anchoring system comprises: anchor bolts, anchoring agent, surrounding rocks, and parts of the anchor bolts. An anchor bolt is divided into exposed, free, and anchorage zones (Fig 1) along its length. When the anchor bolt is subjected to pull-out effects, the axial force in the free zone is transferred to the anchorage zone due to elastic deformation therein. Based on bonding, friction, and mechanical meshing between the anchor bolt and anchoring agent, the circular binding body formed by the anchoring agent, and the effect of the borehole wall, load is transferred to the surrounding rock. The anchoring force refers to the binding force between the anchorage zone of anchor bolts and a rock mass, that is, the constraint force on the anchor bolt from the surrounding rock, which is frequently considered as an important index with which to measure anchor integrity.

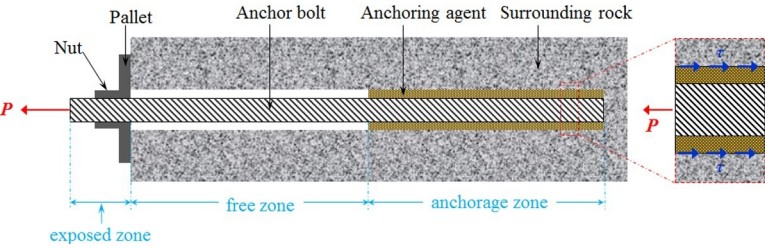

**Fig 1. Force transfer mechanism in an anchor bolt.**

Based on the force transfer process of anchoring system, it can be seen that there are three mechanical interfaces in the anchoring system. When analysing the mechanical properties of the anchorage zone in the elastic stage, the two interfaces (including anchor bolt–anchoring agent and anchoring agent–borehole wall interfaces) were explored. When applying pull-out force to an anchor bolt, the shear stress on the anchorage zone depends on the coupling mechanism between interfaces[24,25]. For grouted anchor bolts, relative displacement occurs between the anchor bolts and surrounding slurry, thus failing in slip on the anchor bolt–anchoring agent interface. Then, the shear stress on the interface is lower than the ultimate shear strength of the interface[26]. For a resin anchor bolt, the anchor bolt is deformed with its anchoring agent, generally failing in slip on the anchoring agent–borehole wall interface[27]. In this case, the shear stress on the interface is equivalent to the ultimate shear strength. The latter was explored in the present study.

According to different deformation forms of anchoring agent–borehole wall interface, the pull-out process of anchor bolts into three stages was simplified[5,28], as shown in Fig 2.

In Stage I (elastic deformation stage), the shear stress is proportional to the shear displacement of the interface which is intact. In this case, $0 \leq \mu \leq \mu_1$ and the relationship between shear stress $\tau$ and displacement $\mu$ is expressed as follows:

$$\tau = \frac{\tau_1}{\mu_1} \mu \tag{1}$$

where, $\tau_1$ and $\mu_1$ refer to the ultimate bonding strength of anchorage body and shear displacement at the ultimate bonding strength of anchorage zone, respectively.

In Stage II (interface softening and damage stage), the interface is partly damaged and therefore shear stress linearly declines with shear displacement. In this context, $\mu_1 \leq \mu \leq \mu_2$ and the shear stress can be calculated as follows:

$$\tau = \frac{\tau_1 - \tau_2}{\mu_1 - \mu_2} \mu + \frac{\tau_2 \mu_1 - \tau_1 \mu_2}{\mu_1 - \mu_2} \tag{2}$$

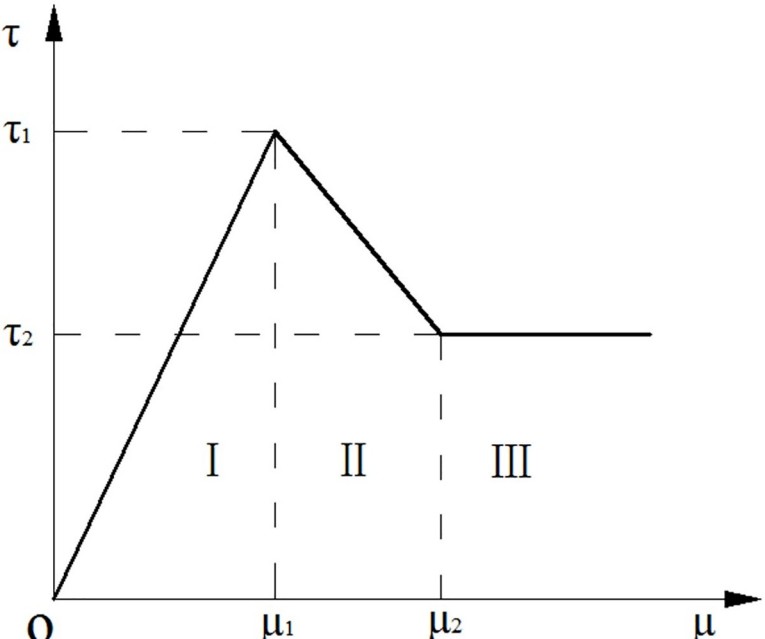

**Fig 2. Shear stress–displacement relationship on the anchoring agent–borehole wall interface.**

where, $\tau_2$ and $\mu_2$ are the residual bonding strength of anchorage zone and the minimum shear displacement under the residual bonding strength of the anchorage zone, respectively.

In Stage III (residual strength stage), the interface was completely damaged; in this context, $\mu \geq \mu_2$ and the shear stress is expressed as follows:

$$\tau = \tau_2 \tag{3}$$

By modifying the micro-element model[29,30], the distribution equation for axial force in the anchorage zone is expressed as follows:

$$P(x) = \frac{e^{\beta x} - e^{\beta(2L_b - x)}}{(1 - e^{2\beta L_b})} P \tag{4}$$

The equation for shear stress distribution of anchoring agent–borehole wall interface is as follows:

$$\tau(x) = \frac{e^{\beta x} + e^{\beta(2L_b - x)}}{\pi D (e^{2\beta L_b} - 1)} \beta P \tag{5}$$

where, $D$, $P$, and $\beta$ separately denote the diameter of the borehole, pull-out force of an anchor bolt, and a material parameter given by:

$$\beta^2 = \frac{4\tau_1}{\mu_1 D E_a} \tag{6}$$

where, $E_a$ is the elastic modulus of the anchorage zone.

$$E_a = \frac{E_b d^2 + E_r(D^2 - d^2)}{D^2} \tag{7}$$

where, $E_b$ is the elastic modulus of bolt, $E_r$ is the elastic modulus of resin anchorage agent, and D is the diameter of bolt. According to Eqs 4 and 5, the distribution curves of axial force and shear stress in the anchorage zone are drawn, as shown in Fig 3.

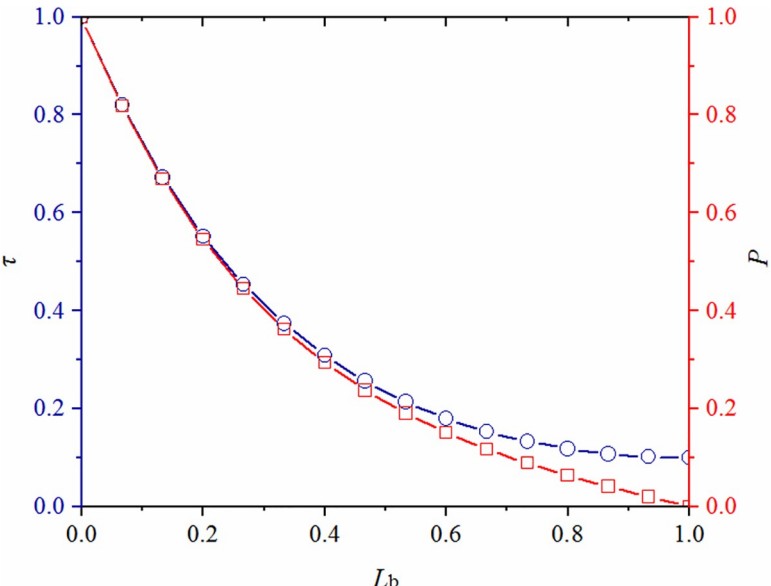

**Fig 3. Distributions of axial force and shear stress in the anchorage zone.**

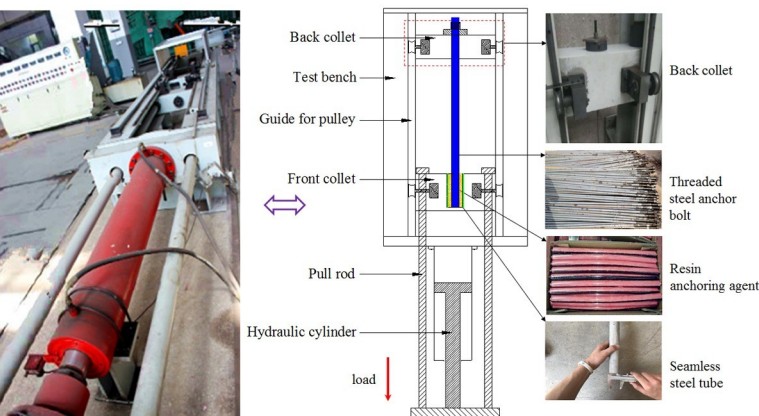

**Fig 4. Apparatus for the pull-out test and test materials.**

The axial force and shear stress of anchorage body monotonically decreased from the beginning to the end of the anchorage zone while the rate of change thereof gradually declined. At the beginning ($x = 0$) of the anchorage zone, the axial force and shear stress on the anchorage body were at a maximum and the axial force was equivalent to that in the free zone of an anchor bolt. On condition of having sufficient pull-out force, relative displacement and damage first appeared at the beginning of the anchorage zone. Afterwards, damage gradually extended to the end of the anchorage zone. At the end ($x = L_b$) of the anchorage zone, the axial force was zero while there was still a residual shear stress present.

## The influence of anchorage length on the stress distribution in the anchorage zone

Bolt support is complex and concealed from observers, so it is hard to measure the deformation and stress on the anchor bolts in field. It is necessary to verify the result obtained through theoretical analysis by conducting laboratory testing and FEM to analyse the load transfer characteristics of an anchoring system.

### Method

**Laboratory test.** In the test, the left-handed threaded steel anchor bolts were applied and the thick-walled steel tube and resin cartridge were separately taken as the anchoring matrix and binding material (Fig 4). Considering the binding effect of this resin anchoring agent, a seamless steel tube with the inner diameter of 30 mm was used, in which threads were processed. The parameters of test materials are shown in Table 1.

The pull-out test was conducted by applying an LW-1000 horizontal tensile test machine (Fig 4). Before the test, the back collet was fixed by using a latch and the end of the anchor bolt with threads was placed into the back collet and fixed through pallet nuts. Moreover, the

**Table 1. Parameters of mechanical properties of the test materials.**

| Anchor bolt | Types of anchor bolts | Diameter/ mm | Length/ mm | Tensile strength/ MPa | Yield strength/ MPa | Breaking force / kN |
|---|---|---|---|---|---|---|
| | Threaded steel | 20 | 2000 | 570 | 400 | 218.7 |
| Anchoring agent | Type | Characteristic | Length/ mm | Diameter/ mm | Gelation time/ s | Waiting time for installation / s |
| | Z2350 | Intermediate speed | 500 | 23 | 91~180 | 480 |

**Table 2. Mechanical parameters of materials.**

| Performance parameters | Tensile strength /MPa | Yield strength/MPa | Shear modulus/GPa | Bulk modulus/GPa | Cohesion /MPa | Internal friction angle /° |
|---|---|---|---|---|---|---|
| Anchoring agent | 15 | - | - | - | - | - |
| Anchor bolt | 570 | 400 | - | - | 12 | 32 |
| Surrounding rocks | 2.1 | 0.96 | 3.3 | 5.1 | 4.6 | 38 |

anchor end (seamless steel tube) was fixed using a front collet. During the test, the front collet was driven through a piston and a pull rod to move away from the back collet to simulate a pull-out force on the anchor bolt. A sensor was used to collect and transfer data (in real time) to a computer.

**Numerical simulation.** A FLAC3D numerical model was established. During simulation, the anchorage interface in a rock mass was simulated by applying interface elements while contact elements were used to simulate the contact interface of media effecting force transfer. The interface elements were used for simulation based on the Mohr-Coulomb model. The contact parameters and deformation characteristics of different interface of anchorage are different. In the process of numerical simulation, the element contact constitutive model was adjusted by setting different contact mechanical parameters(In the anchor bolt–anchoring agent interface, the ultimate shear stress and shear stiffness are 8 MPa and 500 MPa/m. In the anchoring agent–borehole wall interface, the ultimate shear stress and shear stiffness are 4.5 MPa and 300 MPa/m.), in which anchor bolt was simulated by using an isotropic elasticity model. The sandstone with higher strength is chosen as the anchored rock mass, which can better reflect the stress of the anchoring solid in the elastic stage. The sandstone comes from Taoyuan Coal Mine in Anhui Province, China, and its mechanical parameters have been measured in the laboratory, as shown in Table 2.

The model measures 1.0 m × 1.0 m × 1.2 m (length × width × height) and the total length of anchor bolt was 1.2 m, including an anchorage zone and a free zone of 1.0 m and 0.2 m long, respectively. The anchor bolt, with a diameter of 20 mm, was aligned in the centre of the model, with a thickness of anchoring agent of 5 mm simulated.

**Test scheme.** Strain gauges were distributed in the anchorage zone at 100 mm intervals to measure the stress and strain on the anchorage body under the pull-out effect and analyse the change in stress in the anchorage zone. TS3890 static resistance strain gauges were used to measure the strain (Fig 5).

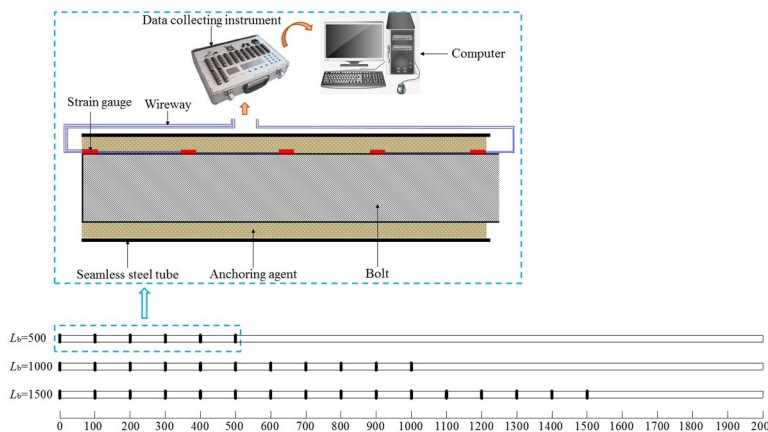

**Fig 5. Distribution and connection of strain gauge.**

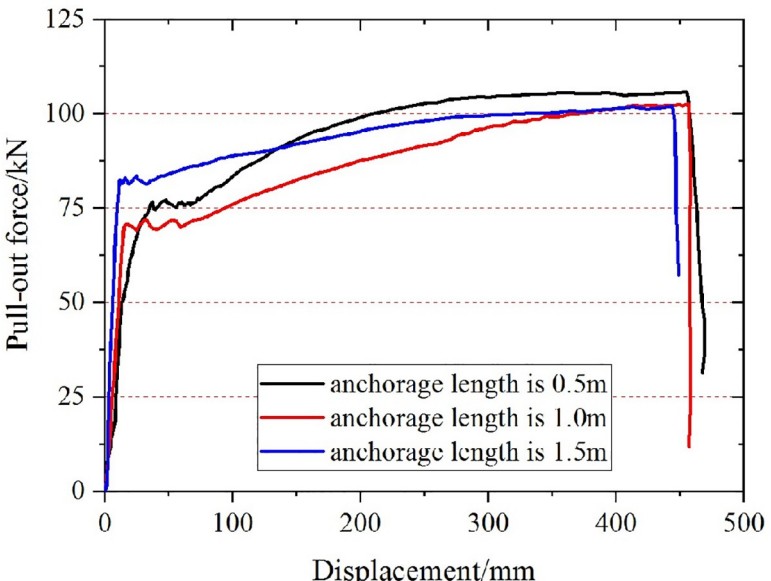

**Fig 6. Relationship between pull-out force and displacement of bolt with different anchorage length.**

During the test, the four-level loads (25, 50, 75, and 100 kN) were separately applied to the anchorage zones with the anchorage lengths of 500, 1000, and 1500 mm. Each scheme was tested twice, so a total of 14 tests were carried out, and then the test data were analyzed and selected. The load was maintained for 3 s and the mechanical response of the anchorage body under different anchorage lengths and pull-out loads analysed.

## Result

**The influence of anchorage length on ultimate pull-out force of anchor bolt.**   As shown in Fig 6, at the initial stage of the test, the pull-out force of the anchor increases rapidly, and then it increases slowly when the anchor reaches the yield strength, but the deformation displacement is large. When the anchorage length is different, the anchor bolt is broken in the free section(Fig 7), not in the anchorage section. The ultimate pull-out force of the anchor bolt is basically the same as the theoretical tensile strength, which is greater than 100kN. It can be seen that the strength of the anchor bolt meets the test requirements of applying the fourth level load on the anchorage body, and during the test, the anchorage section is in the elastic stage (no damage).

**The influence of anchorage length on stress distribution in the anchorage zone.**
(1) Shear stress

Based on measured parameters of anchor bolts for mining service and surrounding rocks, the elastic moduli of the anchorage body and resin cartridge, diameter of anchor bolt, diameter of borehole, and Poisson's ratio of surrounding rocks were 200 GPa, 3 GPa, 20 mm, 30 mm, and 0.24, respectively. On this basis, the curves for comparing changes of shear stress based on laboratory test are shown in Fig 8.

Fig 8 shows the shear stress distributions on interfaces in the anchorage zone for anchorage lengths of 0.5, 1, and 1.5 m when the pull-out force was 50 kN. It can be seen from the Fig 8 that under the same pull-out force and different anchorage lengths, the shear stress on the interfaces did not change linearly but reached a maximum at the beginning of the anchorage zone and gradually reduced to zero with increasing distance from the beginning. The interface

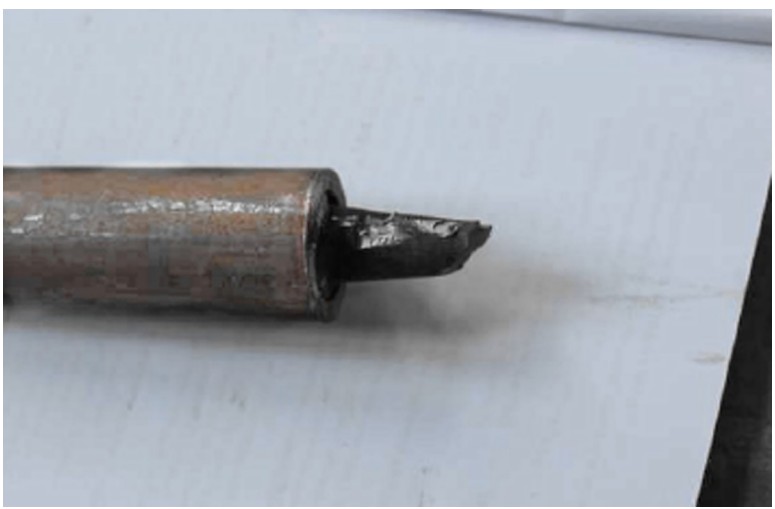

**Fig 7. Failure mode of test bolt.**

was mainly stressed close to the end of the free zone. The shorter the anchorage length, the more uniformly the shear stress was distributed along the anchorage zone and the higher the maximum shear stress on the interfaces. With increasing anchorage zone length, the shear stress on the interfaces decreased and was gradually transferred to the section near the end of

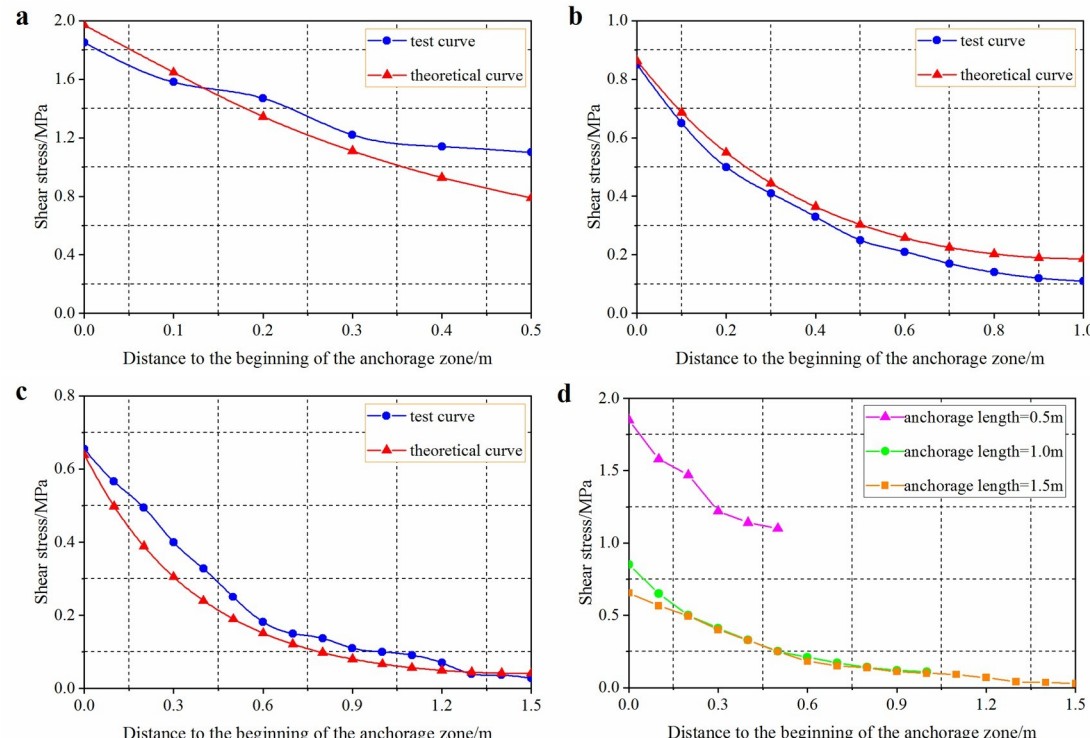

**Fig 8. Shear stress distribution of anchorage body under a same pull-out force and different anchorage lengths.** Anchorage lengths of 0.5 m (a), 1.0 m (b), and 1.5 m (c), (d) is the comparison of shear stress distribution of anchorage body when anchorage length is 0.5m, 1.0m and 1.5m.

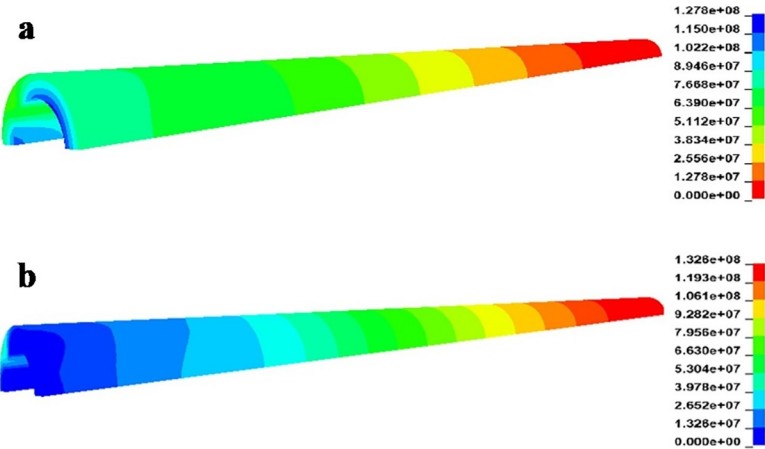

**Fig 9. Stress distribution in the anchoring agent at different anchorage lengths.** (a) anchorage length is 0.5 m; (b) anchorage length is 1.0 m.

the anchorage zone. At the end nearest the applied load (near end, hereinafter), debonding occurred and the shear stress was gradually transformed into a frictional resistance. In this case, the shear stress on the anchorage body was low at a certain distance from the near end. When the anchorage length reached a certain level, the distribution curves of shear stress on interfaces gradually coincided, implying that further increasing the anchorage length had little significant effect on the maximum shear stress.

Fig 9 shows stress distributions in the anchoring agent at anchorage lengths of 0.5 and 1.0 m based on numerical simulation. Shear stress was mainly distributed within a small zone in the near end and shear stress was exponentially distributed and gradually declined from the near end to the far end. The longer the anchorage, the wider the distribution of shear stress and the lower the corresponding shear stress; moreover, the longer the anchorage, the nearer the shear stress was to zero in the anchorage zone. The stress distribution on the anchorage body in the numerical model shows similarities with analytical solutions based on the shear–slip model. In engineering practice, it is necessary to reinforce the vicinity of the interface as much as possible to guarantee the strength of surrounding rocks near the interface and also ensure the integrity of anchorage in the initial segment.

(2) Analysis of axial stress

The axial stress is given by:

$$\sigma_i = \varepsilon_i E_s \tag{8}$$

where, $\sigma_i$ and $\varepsilon_i$ denote the axial force and strain at point $i$, respectively.

The axial force at the borehole mouth was equivalent to that in the free zone. With a resin anchoring agent, the axial force distribution varied and was different from the equivalent distribution in the free zone. The axial stress gradually decreased from the outer end to the tailing end of the anchor because the cohesion at the near end of the anchor bolt was gradually overcome with increasing pull-out load and the interface at the tailing end was constantly driven to resist the pull-out load. Additionally, the axial stress of anchor bolt correspondingly increased. The comparison results of theoretical analysis, laboratory test and numerical simulation are as follows.

As shown in Fig 10, when applying a pull-out force of 50 kN, the axial force varied quasi-linearly when the anchorage length was 0.5 m. With increasing anchorage length, the axial force of anchor bolts became less uniform. When the anchorage length was 1500 mm, the axial

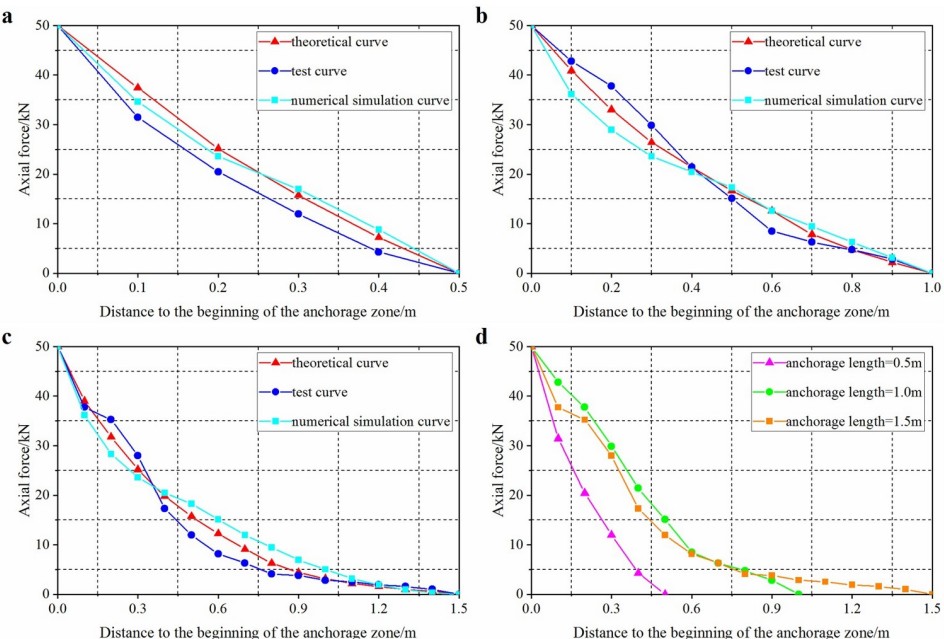

**Fig 10. Axial stress distributions in anchorage zone with different anchorage lengths and a given pull-out force.** Anchorage lengths of 0.5 m (a), 1.0 m (b), and 1.5 m (c), (d) is test curves of three length.

force was mainly distributed in the vicinity of the borehole mouth and decreased with distance therefrom. At a certain anchorage length, the axial force tended to zero and the peak axial force was unaffected; however, due to the increase in anchorage length, the zone over which the axial force was distributed expanded and therefore the anchor bolt further from the anchorage interface was subjected to a small axial force. That is, it exhibited sufficient bearing capacity and can thus bear more load. The result obtained through numerical simulation was consistent with that obtained by analytic calculation.

## The influence of pull-out force on the stress distribution in the anchorage zone

(1)Distribution of axial stress under different pull-out forces

When the anchorage length was 1.0 m, the changes in axial stress of anchorage zone under three-level pull-out forces (25, 50, and 75 kN) were simulated. In Fig 11, the axial force is seen to be non-linearly distributed along the anchor. In the elastic stage, anchor bolts showed the same trend of stress distribution with increasing load, moreover, stress changes were mainly found at the beginning of the anchorage zone where the ultimate pull-out force was first mobilised. On this basis, it can be inferred that the anchorage body of an anchor bolt was first damaged at the beginning of its anchorage zone.

(2)Distribution of shear stress under different pull-out forces

Under low load, the interface between the anchoring agent and the anchor bolt at the borehole mouth was subjected to elastic deformation. In this case, the anchorage body was undamaged and shear stress within the anchorage zone gradually reduced and was uniformly distributed. With increasing load, the shear stress rapidly rose to its peak within a short distance from the borehole mouth: this implied that shear failure started to occur at the beginning of the anchorage zone and the failure gradually extended to the deeper anchorage interface with increasing load. As the maximum shear stress remained unchanged, the locus of the peak

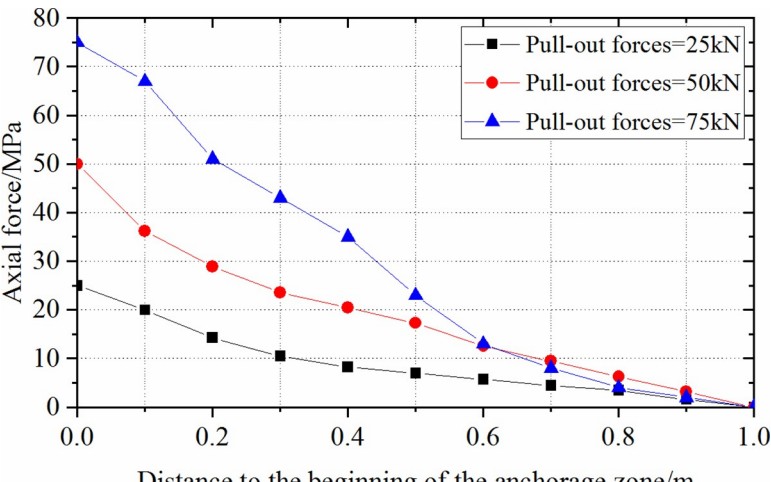

**Fig 11. Axial force distributions of anchorage zone under different pull-out forces in laboratory test.**

shear stress shifted to the deeper anchorage zone. With a large anchorage length, there was a wider response range to external load within the anchorage zone, so the anchorage body can bear a larger load, thus improving the bearing capacity of the anchorage zone. By analysing Fig 12, it can be found that, within the ultimate bearing range, the larger the pull-out force, the less uniform the stress distribution; the longer the anchorage, the more centralised the shear stress on the interface at the beginning of the anchorage zone.

## Determination of reasonable anchorage length

It can be seen from Fig 8(D) and Fig 10(D) that there was a critical length of anchorage zone under the effect of pull-out force, beyond which the ultimate bearing capacity of the anchor bolts did not increase. When the external load reached a certain level, the anchorage layer changed from one undergoing elastic deformation to elasto-plastic deformation and the shear stress on the anchorage interface did not continue to increase. To guarantee anchorage body function, the maximum shear stress on the anchorage zone cannot exceed the ultimate shear strength of the anchorage body–rock interface, which was taken as the main controlling condition for determining the anchorage length. In this context, the resistance at the beginning of the anchorage zone was equivalent to the ultimate shear stress $[\tau]$ on the interface. By

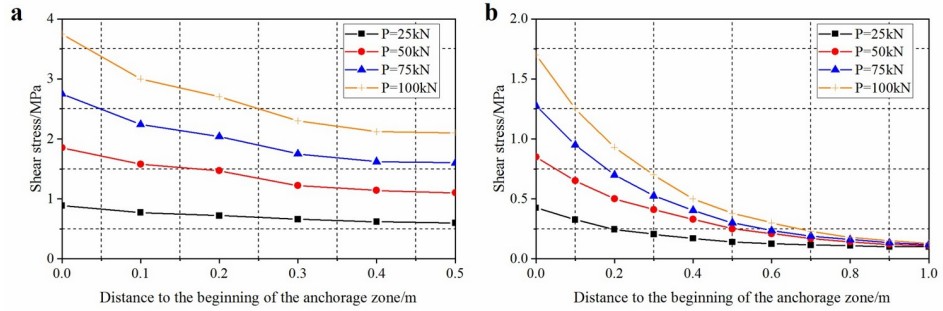

**Fig 12. Shear stress distributions of anchorage zone under different pull-out forces in laboratory test.** (a) anchorage length is 0.5 m; (b) anchorage length is 1.0 m.

simultaneously using Eq 4, the ultimate pull-out force of the anchorage zone can be obtained thus:

$$P_{\max} = \frac{\pi D[\tau](e^{2\beta L_b} - 1)}{\beta(1 + e^{2\beta L_b})} \qquad (9)$$

Owing to $\tanh x = \frac{e^x - e^{-x}}{e^x + e^{-x}}$, assuming $x = \beta L_b$, the following result can be attained:

$$P_{\max} = \frac{\pi D[\tau]}{\beta}\tanh(\beta L_b) \qquad (10)$$

The ultimate bearing capacity of anchoring system increased with increasing anchorage length and shear capacity of the anchorage interface. With the constant growth of anchorage length, the bearing capacity of the anchoring system increased, then stabilised, as shown in Fig 13.

When $\beta L_b$ was infinite, $\tanh(\beta L_b)$ tended to unity; however, in practical engineering, it not only needs to be technically satisfactory, but also cost-effective. According to the peak, and incremental, axial force, the eigenvalues of the system can be attained (Table 3).

According to the corresponding relationship between $P_{\max}$ and $\beta L_b$ in Table 3, it can be seen that the increment of $\beta L_b$ increased with $P_{\max}$. This meant that, after reaching a certain critical value, the anchorage length needs to be increased by much more when augmenting the axial force on the anchor bolt by the same amount. Therefore, there is a certain reasonable length range, in which technical and economic effects can both be satisfied. When $P_{\max} > 0.9$, it is supposed that $k$ denotes the increment of $\beta L_b$ required for the same increase in axial force on the anchor bolt, that is, the efficiency of increasing the peak axial force of anchor bolt by increasing the anchorage length (Fig 14) can be deduced.

As shown in Fig 14, when $P_{\max} < 0.98$, the increment in $\beta L_b$ and $k$ increased slightly; when $P_{\max} \geq 0.98$, the increment in $\beta L_b$ and $k$ both increased, therefore, $P_{\max} = 0.98$ can be considered as a criterion for discriminating a reasonable anchorage length, with which economic principles are also satisfied on the premise of realising the desired technical end. In this case, $\beta L_b = 2.3$, so the reasonable anchorage length of such anchor bolts was $0.435\beta$, that is, $0.87\sqrt{\frac{\tau_1}{\mu_1 D E_a}}$.

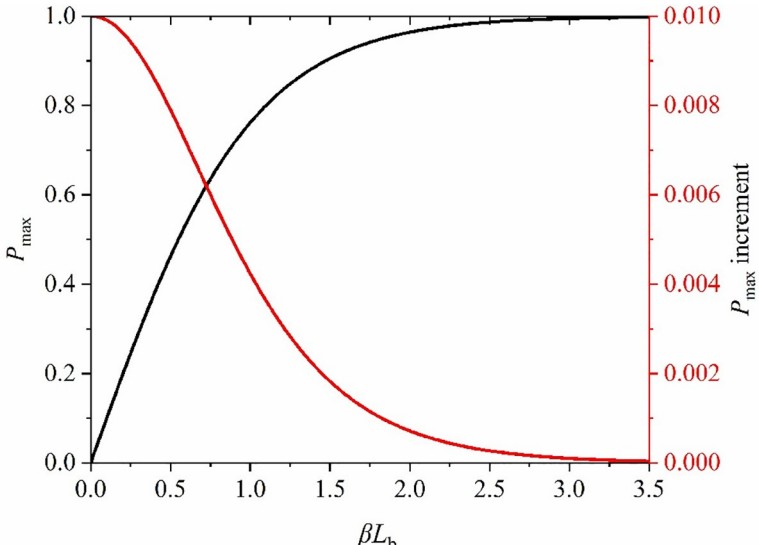

**Fig 13. Changes of peak value and incremental of axial force in anchorage zone along with its length.**

**Table 3. A comparison between the peak axial force and βL_b eigenvalues.**

| $P_{max}$ | 0.9 | 0.91 | 0.92 | 0.93 | 0.94 | 0.95 | 0.96 | 0.97 | 0.98 | 0.99 | 0.995 |
|---|---|---|---|---|---|---|---|---|---|---|---|
| $\beta L_b$ | | 1.48 | 1.53 | 1.59 | 1.66 | 1.74 | 1.84 | 1.95 | 2.1 | 2.3 | 2.65 | 3 |
| $P_{max}$ increment / $\times 10^{-5}$ | | 189 | 173 | 155 | 136 | 117 | 97 | 79 | 59 | 40 | 20 | 10 |

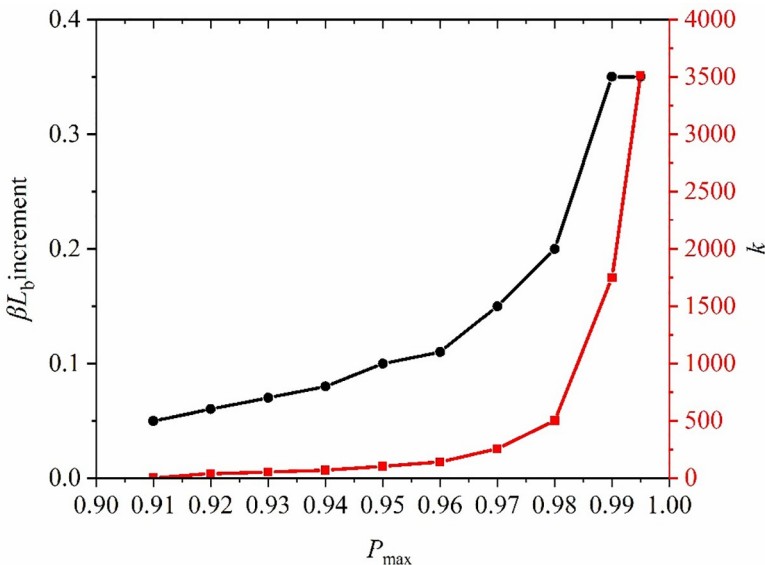

**Fig 14. Changes of $k$ and the increment of $\beta L_b$ along with the peak value of axial force.**

## Conclusions

(1) Based on the shear–displacement model, the analytical expressions for the distribution of axial force on the anchorage body and shear stress on the anchorage body–surrounding rock interface along the anchorage zone were attained. Furthermore, based on the shear–displacement model, it was found that the axial force decreased in a non-uniform manner along the anchor bolt to the deeper anchorage zone. Moreover, the shear stress on interface at the beginning of anchorage zone of the anchor bolts was maximised, then decreased along anchor.

(2) The influence of anchorage length on the stress distribution along an anchor bolt was obtained: in the elastic deformation stage, the longer the anchorage length, the more uniform the shear stress distribution along the anchorage zone and the higher the maximum shear stress on the interface. Beyond a certain critical anchorage length, further increases therein caused no significant influence on the maximum shear stress.

(3) It was shown that there was a critical anchorage length: as the peak axial force on the anchor bolts exhibited a hyperbolic tangent relationship with the anchorage length, it was determined that the technical and economic effects of an anchor bolt support system can be realised when the optimal anchorage length was $0.435\beta$.

## Supporting information

**S1 File. Figure data.**
(XLSX)

**S1 Table. Parameters of mechanical properties of the test materials.**
(DOC)

**S2 Table. Mechanical parameters of materials.**
(DOC)

**S3 Table. A comparison between the peak axial force and $\beta L_b$ eigenvalues.**
(DOC)

## Acknowledgments

This work was supported, and financed, by the General Program of the National Natural Science Foundation of China (51864044). The authors would like to thank the Editor and the Reviewers for their helpful and constructive comments.

## Author Contributions

**Conceptualization:** Xingliang Xu.

**Data curation:** Suchuan Tian.

**Validation:** Xingliang Xu, Suchuan Tian.

**Writing – original draft:** Xingliang Xu.

**Writing – review & editing:** Xingliang Xu, Suchuan Tian.

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
