## [Decision Letter · Decision Letter 0]

5 Sep 2019

PONE-D-19-21977

Load transfer mechanism and critical length of anchorage zone for anchor bolt

PLOS ONE

Dear Dr Tian,

Thank you for submitting your manuscript to PLOS ONE. After careful consideration, we feel that it has merit but does not fully meet PLOS ONE’s publication criteria as it currently stands. Therefore, we invite you to submit a revised version of the manuscript that addresses the points raised during the review process.

We would appreciate receiving your revised manuscript by Oct 20 2019 11:59PM. To enhance the reproducibility of your results, we recommend that if applicable you deposit your laboratory protocols in protocols.io, where a protocol can be assigned its own identifier (DOI) such that it can be cited independently in the future. For instructions see: http://journals.plos.org/plosone/s/submission-guidelines#loc-laboratory-protocols

We look forward to receiving your revised manuscript.

Kind regards,

Anna Pandolfi, Ph.D.

Academic Editor

PLOS ONE

Journal Requirements:

1. Please include in your Methods section the vendor details for all equipment and materials used, in order to enhance reproducibility.

2. We note you have included a table to which you do not refer in the text of your manuscript. Please ensure that you refer to Table 2 in your text; if accepted, production will need this reference to link the reader to the Table.

Additional Editor Comments (if provided):

Please answer the comments of the reviewers and try to improve the general presentation of the manuscript.

Reviewers' comments:

Reviewer's Responses to Questions

**Comments to the Author**

1. Is the manuscript technically sound, and do the data support the conclusions?

Reviewer #1: Partly

2. Has the statistical analysis been performed appropriately and rigorously? 

Reviewer #1: N/A

3. Have the authors made all data underlying the findings in their manuscript fully available?

Reviewer #1: No

4. Is the manuscript presented in an intelligible fashion and written in standard English?

Reviewer #1: Yes

5. Review Comments to the Author

Reviewer #1: The response of resin grouted anchors with the steel threaded tendon to the axial loading is explored in presented study via laboratory experiments and numerical simulations. The simplified trilinear analytical model is used for mathematical description of the relationship between the shear stress and displacement in the bond length of anchor. The equation for the deduction of critical anchor length was determined based on the analytical model supported by the experimental and numerical results.

I agree with the authors note, which says: “The load transfer mechanism of anchor bolts is a research hot-spot “. The presented manuscript has a great chance to become a valuable contribution to the discussion regarding the determination of both safe and cost-effective rock anchors. However, the manuscript needs to undergo major revision before it will be published. See the general comments to the manuscript below. (The specific questions, comments and notes are added in the attached pdf file with manuscript).

General comments:

- The laboratory testing campaign was performed. At least some summary of the number of conducted tests (with the description of every group of tested anchors regarding lengths) should be presented. Otherwise the support of the conclusions by the experiments can´t be evaluated properly.

- The analytical model which was used is not applicable without the knowledge of shear stress and shear displacement values related to the specific anchor. Stress-strain diagram of tested anchors (Or at least the values of stress and strain which are essential for the use of mentioned model) is worth to include in article. It is in line with the demand of availability of the data underlying the findings.

- The more attention should be paid to the logical organizing structure of text. There are some results presented in the paragraph 3.3 (Results). Then the methodology of numerical modelling is mentioned (Methods). From this point in the manuscript it can starts to be complicated to the reader to distinguish which discussed results are related to the laboratory experiment and which to the numerical simulation. It would be also good to improve the captions of graphs in this way.

6. PLOS authors have the option to publish the peer review history of their article (what does this mean?). If published, this will include your full peer review and any attached files.

Reviewer #1: No

---

## [Author Response · Author response to Decision Letter 0]

24 Oct 2019

Dear Editor:

Thank you very much for your letter on 6 September, 2019, regarding the comments and suggestions made on our paper entitled “Load transfer mechanism and critical length of anchorage zone for anchor bolt” (ID: PONE-D-19-21977). The comments are very valuable and very helpful to us when revising and improving our manuscript, as well as the important guiding significance to our researches. We have studied the comments carefully and have made the required revisions. 

Enclosed are a clean copy of our revised manuscript (see ‘Manuscript-Revised’), a marked-up copy of our manuscript that highlights changes made to the original version (see ‘Revised Manuscript with Track Changes’ and ‘Manuscript-Revised _HighlightedCopy’), and a detailed response letter to the editors and reviewers explaining the changes we have made (see ‘Response to the Reviewers’).

The main revisions to the manuscript and our response to you and the reviewers’ comments are as follows:

Journal Requirement:

1. Please include in your Methods section the vendor details for all equipment and materials used, in order to enhance reproducibility.

Response: Thank the Editor for the kind recommendation. The LW-1000 horizontal tensile test machine is provided by Hangzhou Yingmin Technology Co., Ltd(Zhejiang China). The left-handed threaded steel anchor bolts and the resin cartridge are provided by Xuzhou Baoding Support Technology Co., Ltd(Jiangsu China). The strain gauges and TS3890 static resistance strain gauges are provided by Jiangsu Donghua Testing Technology Co., Ltd(Jiangsu China).

2. We note you have included a table to which you do not refer in the text of your manuscript. Please ensure that you refer to Table 2 in your text; if accepted, production will need this reference to link the reader to the Table.

Response: We are very sorry for our negligence and thanks for the reviewer’s kind suggestion. We have described and referenced Table 2 in the text. The revised details can be found in Line 130-132, Page 5(Revised Manuscript).

Reviewer Comment:

1. Is the manuscript technically sound, and do the data support the conclusions?

Response: Thanks for the reviewer. The conclusions of this paper are obtained by theoretical analysis, laboratory test and numerical simulation. The data of this paper are all real data. The result of data analysis is promoted to the theoretical level, so as to reach the final conclusion, which is in line with the logic thinking and methods commonly used in scientific research. Moreover, the operation process of the laboratory test in this paper conforms to the test specifications, so the data obtained is reliable enough to support the conclusion of this paper.

2. Has the statistical analysis been performed appropriately and rigorously?

Response: Thanks for the reviewer.All the data in this paper are real data. The analysis of the data was carried out according to the corresponding scientific norms without any falsification. The conclusions of this paper are based on the real data and scientific and technological standards, and the conclusions are credible. Therefore, the author thinks that the statistical analysis is appropriate and rigorous.

3. Have the authors made all data underlying the findings in their manuscript fully available?

Response: Thanks for the reviewer’s kind suggestion. The curves in this paper are all based on the obtained data, which can reflect the data and its changing trend. If the data supporting all graphs is added to the paper, it will not only duplicate the contents of graphs, but also lead to the verbose and complex structure of the paper. Therefore, all data are provided separately. Please refer to attached files for details.

4. Is the manuscript presented in an intelligible fashion and written in standard English?

Response: Thank the Reviewer for the kind recommendation. The English throughout has been revised by the native English-speaking professors using track change in MS Word. ‘Language Revision with Track Change.doc’ and ‘Language Revision Certificate.pdf’ issued by the service, are all uploaded for your references.

5. ①The laboratory testing campaign was performed. At least some summary of the number of conducted tests (with the description of every group of tested anchors regarding lengths) should be presented. Otherwise the support of the conclusions by the experiments can´t be evaluated properly.

Response: Thanks for the reviewer’s kind suggestion. In this paper, the mechanical properties of the anchorage body of the anchor under different anchorage lengths (500mm、1000mm and 1500mm) and pull-out forces (25kN、50kN、75kN、100kN) are tested. Each scheme was tested twice, so a total of 14 tests were carried out, and then the test data were analyzed and selected. We add the above summary to the paper. The revised details can be found in Line 149-150 and Page 6.

②The analytical model which was used is not applicable without the knowledge of shear stress and shear displacement values related to the specific anchor. Stress-strain diagram of tested anchors (Or at least the values of stress and strain which are essential for the use of mentioned model) is worth to include in article. It is in line with the demand of availability of the data underlying the findings.

Response: Thanks for the reviewer’s kind suggestion. This paper focuses on the mechanical characteristics of the anchorage section in the elastic stage. According to the previous studies (reference 29, 30, etc.), the mechanical model adopted in this paper is the most consistent with the shear stress displacement relationship of the anchorage interface in the elastic stage, and the experimental research in the following paper also verifies the correctness of the model. At the same time, according to the suggestions put forward by the reviewer, the relationship diagram between pull-out force and displacement and failure form diagram of the anchor during the test process were added, which provides more evidence for the credibility of the data on which the research results are based. The revised details can be found in Line 153-160, page 4.

③The more attention should be paid to the logical organizing structure of text. There are some results presented in the paragraph 3.3 (Results). Then the methodology of numerical modelling is mentioned (Methods). From this point in the manuscript it can starts to be complicated to the reader to distinguish which discussed results are related to the laboratory experiment and which to the numerical simulation. It would be also good to improve the captions of graphs in this way.

Response: Thanks for the Reviewer. According to the reviewer's suggestion, the author has reorganized the article structure of the third part in the form of research method-research scheme-research results. Please see the new manuscript for details. The revised details can be found in Line 116-117,129-143,161-162,187,205-206,213, page 5-8.

We would like to thank the Reviewers for the kind comments and appreciate the Editor’s and Reviewers’ diligence. We trust that the revisions will meet with your approval.

Once again, thank you very much for your time in this matter.

Yours sincerely,

Xingliang Xu, Suchuan Tian

---

## [Decision Letter · Decision Letter 1]

15 Nov 2019

PONE-D-19-21977R1

Load transfer mechanism and critical length of anchorage zone for anchor bolt

PLOS ONE

Dear Dr Tian,

Thank you for submitting your manuscript to PLOS ONE. After careful consideration, we feel that it has merit but does not fully meet PLOS ONE’s publication criteria as it currently stands. Therefore, we invite you to submit a revised version of the manuscript that addresses the points raised during the review process.

We would appreciate receiving your revised manuscript by Dec 30 2019 11:59PM. To enhance the reproducibility of your results, we recommend that if applicable you deposit your laboratory protocols in protocols.io, where a protocol can be assigned its own identifier (DOI) such that it can be cited independently in the future. For instructions see: http://journals.plos.org/plosone/s/submission-guidelines#loc-laboratory-protocols

We look forward to receiving your revised manuscript.

Kind regards,

Anna Pandolfi, Ph.D.

Academic Editor

PLOS ONE

Additional Editor Comments (if provided):

You are invited to address both comments of the reviewer, with a particular care on the first one.

Reviewers' comments:

Reviewer's Responses to Questions

**Comments to the Author**

1. If the authors have adequately addressed your comments raised in a previous round of review and you feel that this manuscript is now acceptable for publication, you may indicate that here to bypass the “Comments to the Author” section, enter your conflict of interest statement in the “Confidential to Editor” section, and submit your "Accept" recommendation.

Reviewer #1: (No Response)

2. Is the manuscript technically sound, and do the data support the conclusions?

Reviewer #1: Yes

3. Has the statistical analysis been performed appropriately and rigorously? 

Reviewer #1: Yes

4. Have the authors made all data underlying the findings in their manuscript fully available?

Reviewer #1: Yes

5. Is the manuscript presented in an intelligible fashion and written in standard English?

Reviewer #1: Yes

6. Review Comments to the Author

Reviewer #1: The calculation formula for the critical anchorage length of the bolt is derived in the article. The plausibility of presented equation is supported by result of laboratory tests performed on ground anchors and by the results of FEM numerical modeling.

Comments to the revised manuscript:

The structure of the text of reviewed version is much better organized. Text is now logically sequenced. Many missing information was added. Misunderstandings were clarified. However, there are still two more things needed to be done, from my point of view. The first commend arose after the author´s revision of the manuscript. The second comment has not been addressed by the authors.

1/ The term “different” used in the recently added lines 134-136 need to be explained. I cite: …” … deformation characteristics of different interface of anchorage are different”.” … model was adjusted by setting different contact mechanical parameters”. The values of parameters mentioned in these sentences should be added into the manuscript.

2/ Some extra details about the sensors used for laboratory testing has been introduced by the authors. However, even with this new information, I am still asking one question to the authors: “How were the strain gauges (e. g. 15 gauges in case of Lb = 1500 mm) connected to the datalogger? If a real photo of rod with installed gauges (and/or detail of installation of gauge on the threaded bar) exists, I encourage the authors to publish it, in order to enhance reproducibility of their work. (It can be added, for example, to the fig. 5).

7. PLOS authors have the option to publish the peer review history of their article (what does this mean?). If published, this will include your full peer review and any attached files.

Reviewer #1: No

---

## [Author Response · Author response to Decision Letter 1]

13 Dec 2019

Dear Editor:

Thank you very much for your letter on 15 November, 2019, regarding the comments and suggestions made on our paper entitled “Load transfer mechanism and critical length of anchorage zone for anchor bolt” (ID: PONE-D-19-21977 R1). The comments are very valuable and very helpful to us when revising and improving our manuscript, as well as the important guiding significance to our researches. We have studied the comments carefully and have made the required revisions. 

Enclosed are a clean copy of our revised manuscript (see ‘Revised Manuscript’), a copy of revised manuscript with track changes (see ‘Revised Manuscript with Track Changes’), and a detailed response letter to the editors and reviewers explaining the changes we have made (see ‘Response to Reviewers’).

The main revisions to the manuscript and our response to the reviewers’ comments are as follows:

1. The term “different” used in the recently added lines 134-136 need to be explained. I cite: …” … deformation characteristics of different interface of anchorage are different”.” … model was adjusted by setting different contact mechanical parameters”. The values of parameters mentioned in these sentences should be added into the manuscript. 

Response: Thanks for the reviewer for the good comment. I have added two interface parameters mentioned by reviewers in the new manuscript, namely, "In the anchor bolt–anchoring agent interface, the ultimate shear stress and shear stiffness are 8 MPa and 500 MPa/m. In the anchoring agent–borehole wall interface, the ultimate shear stress and shear stiffness are 4.5 MPa and 300 MPa/m." The revised details can be found in Line 135-137 and Page 5.

2. Some extra details about the sensors used for laboratory testing has been introduced by the authors. However, even with this new information, I am still asking one question to the authors: “How were the strain gauges (e. g. 15 gauges in case of Lb = 1500 mm) connected to the datalogger? If a real photo of rod with installed gauges (and/or detail of installation of gauge on the threaded bar) exists, I encourage the authors to publish it, in order to enhance reproducibility of their work. (It can be added, for example, to the fig. 5).

Response: Thanks for the Reviewer. The wire is connected with one end of the strain gauge. The strain gauge is pasted on the corresponding position of the anchor rod with glue. The wire is led out from the near end of the steel pipe and connected with the data collecting instrument. Place the bolt with strain gauge in the center of the steel pipe, and quickly inject the mixed anchoring agent. The test can be carried out after the anchoring agent solidifies. Due to the author's negligence, some test scenes were not photographed during the test. In order to make up for the author's negligence and provide the following scholars with experimental reference, the author has made a schematic diagram of the installation structure of anchor bolt and strain gauge. The revised details can be found in page 56 in “Revised Figures.doc”.

We would like to thank the Reviewers for the kind comments and appreciate the Editor’s and Reviewers’ diligence. We trust that the revisions will meet with your approval.

Once again, thank you very much for your time in this matter.

Thank you and best regards.

Yours sincerely,

Xingliang Xu, Suchuan Tian

tiansc@cumt.edu.cn

---

## [Editor Report · Decision Letter 2]

23 Dec 2019

Load transfer mechanism and critical length of anchorage zone for anchor bolt

PONE-D-19-21977R2

Dear Dr. Tian,

We are pleased to inform you that your manuscript has been judged scientifically suitable for publication and will be formally accepted for publication once it complies with all outstanding technical requirements.

With kind regards,

Anna Pandolfi, Ph.D.

Academic Editor

PLOS ONE
---

## [Editor Report · Acceptance letter]

6 Jan 2020

PONE-D-19-21977R2 

Load transfer mechanism and critical length of anchorage zone for anchor bolt 

Dear Dr. Tian:

I am pleased to inform you that your manuscript has been deemed suitable for publication in PLOS ONE. Congratulations! Your manuscript is now with our production department. 

With kind regards,

on behalf of

Dr. Anna Pandolfi 

Academic Editor

PLOS ONE